# Study on optimization measures for water level fluctuation of large water conveyance aqueducts

**Jian Chen**[1], **Yangyang Tian**[1], **Huijie Zhang**[1], **Hongling Shi**[2]*

1 Water Conservancy College, North china University of Water Resources and Electric Power, Zhengzhou, China, 2 China Institute of Water Resources and Hydropower Research, Beijing, China

* tyy258362@163.com

## Abstract

This paper addresses the issue of water surface fluctuations in aqueducts caused by the Karman Vortex Street phenomenon, which significantly impacts the structural stability and water delivery efficiency of the aqueducts. Based upon existing research findings, a representative three-dimensional fluid dynamics model is developed to optimize the transition section and tail pier structure of the aqueduct, with the objective of reducing water level fluctuations and improving hydraulic stability. The research focuses on improving the symmetry of both the transition section and the tail pier, optimizing the tail pier structure, and analyzing its wave attenuation effect. The experimental results demonstrate that a conical tail pier performs better than a platform-shaped tail pier in eliminating water surface fluctuations, significantly reducing turbulent energy dissipation at the exit transition section, maintaining a relatively stable Froude number, and achieving a stable flow pattern. Furthermore, by comparing the total head loss at the outlet tapering section, it is found that a shorter (e.g.,15-meter) conical tail pier results in lower head loss than a longer (e.g.,35-meter) conical tail pier, indicating that a compact conical tail pier is more effective. This suggests that shorter tail piers are generally more effective in reducing head loss and improving flow stability. These findings offer valuable insights for optimizing tail pier selection in aqueduct design. This study highlights the crucial role of tail pier structure in reducing water surface fluctuations caused by the Karman Vortex Street phenomenon, thereby providing theoretical support and practical guidance for improving the structural stability and water delivery efficiency of aqueducts.

## 1. Introduction

The Karman Vortex Street was first described by von Kármán as a phenomenon where "two parallel rows of vortices alternating in direction, are shed in the wake of an obstacle in a fluid with moderate Reynolds numbers."[1] The wind-induced damage failure of the Tacoma Narrows Bridge in the United States was an accident caused by the Karman Vortex Street. The shedding frequency of the Karman Vortex Street was close to the natural frequency of the bridge deck, leading to resonance. As a result, the amplitude of the vibrations gradually increased, ultimately causing structural instability and collapse[2]. The alternating lateral

**Data availability statement:** All relevant data are within the manuscript and its Supporting Information files.

**Funding:** The paper is supported by the National Natural Science Foundation of China(Grant No.U22A20237) and the Open Research Fund of Key Laboratory of Sediment Science and Northern River Training, the Ministry of Water Resources, China Institute of Water Resources and Hydropower Research(Grant No. IWHR-SEDI-202103). The funders had no role in study design, data collection and analysis, decision to publish, or preparation of the manuscript.

**Competing interests:** The authors have declared that no competing interests exist.

force perpendicular to the flow direction under the bridge forces the bridge to vibrate. When the release frequency is coupled with the natural frequency of the bridge structure, resonance will occur, causing damage[3]. The Karman Vortex Street is not only observed in bridge structures but also in large-scale water conveyance projects. In large-scale water conservancy hubs or river-crossing bridges, bridge piers serve as crucial support structures and are frequently subjected to the scouring and flow-around effects of water currents. When water flows around a pier at a certain velocity, if the shape, size of the pier, and the water flow velocity meet the conditions for the formation of Karman Vortex Streets, vortices with opposite rotational directions will alternately generate on both sides of the pier, forming Karman Vortex Streets. This phenomenon not only leads to localized turbulence in the water flow around the pier but may also induce vibrations and fatigue damage to the pier[4,5]. In practical engineering, engineers often adopt measures such as optimizing the shape of the pier, adding dampers, or altering the water flow velocity to suppress the generation of Karman Vortex Streets, ensuring the safety and stability of the bridge[6,7]. For example, the Central Route of South-to-North Water Diversion draws water from the Danjiangkou Reservoir, passes through Henan and Hubei provinces, crosses the Yellow River, and supplies water northward to Beijing and other northern regions. It is a major water transfer project connecting southern and northern China, in April 2020, the Central Line Project of South-to-North Water Diversion first delivered water with a design maximum flow of 420m³/s.

As shown in Fig 1, the aqueduct body will have water surface fluctuations, while local areas will also have slapping against support beams, side wall overflow and other phenomena. This phenomenon compromises flow stability within the aqueduct, leading to reduced discharge capacity and decreased overall conveyance efficiency.

In large-scale water conveyance aqueducts, fluctuations in water level not only pose safety concerns but also limit discharge capacity. Some researchers have studied the Karman Vortex Street generated by the tail pier of the aqueduct and come up with improvement measures.A tail pier is a structural component situated at the downstream end of a hydraulic system, such as an aqueduct, dam, or spillway, designed to stabilize and guide water flow.

There have been relevant studies on water conveyance engineering in practice.Li Yijia et al.[8] established a numerical simulation model to simulate the hydraulic characteristics of the open channel transmission system under the regulation of the control gate, and found that the opening and closing rate of the gate was positively correlated with the water level fluctuation;Wu Yongyan et al.[9] took the west trunk canal of a water diversion project in Xinjiang as an example and summarised the flow pattern changes in the transition section from the trapezoidal open channel to the inlet of the horseshoe tunnel through physical model experiments, and analysed the effects of water surface contraction angle and Reynolds number

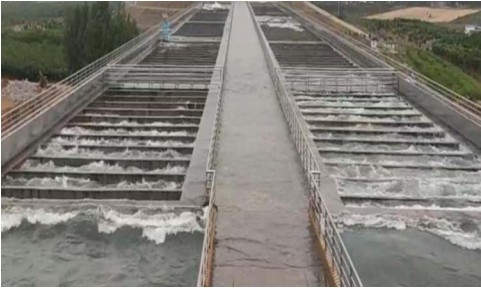

**Fig 1. Fluctuation of water level in aqueduct body.**

on local head loss in the transition section.Wang Caihuan et al.[10], using the method of physical model experiments, clarified that the Karman Vortex Street generated by the outlet flow of the 12li River aqueduct is the root cause of the aqueduct water surface fluctuation. [11] Wang Songtao[12] further studied the water surface fluctuation of the aqueduct on the basis of revealing the mechanism of the water surface fluctuation of the aqueduct body. Wu Zhenghou et al.[13] carry out CFD(Computational Fluid Dynamics) numerical simulations of the hydraulic characteristics of the gradual change section of different forms of crossings. The CFD numerical simulations were carried out using the fluid dynamics software Fluent for the common forms of crossings in practical engineering. Through the analysis, a more suitable form of ferry taper is compared, providing technical support for the design, construction, material selection and maintenance of ferries in actual projects. Li Daming et al.[14]used Flow-3D software to simulate the flow pattern of the RNG (Re- normalization group) turbulence model to achieve similar results to those obtained from physical model experiments.The scheme of lengthening tail pier to reduce water surface fluctuation is put forward, and several types and lengths of vertical pier are studied, which has played a good role in reducing water surface fluctuation.When the tail of the mid pier produces a Karman Vortex Street, the fluid will have a fixed period of alternating transverse forces on the object, and when it is similar to the inherent frequency of the object, it will cause resonance and even damage to the object.

Besides the practical application studies in water conveyance engineering, theoretical analysis of open channel turbulence also provides an important foundation for understanding the flow characteristics of aqueducts. The anisotropy of open channel turbulence and the impact of free surfaces on turbulence intensity are directly related to the flow stability in the transition sections of aqueducts. Therefore, it is necessary to combine the research findings on open channel turbulence to further explore the mechanisms of flow fluctuations in aqueducts.

The anisotropy of turbulence in open channels is caused by the combined effects of the solid boundary and the water surface. The near-wall region of the open channel is the turbulence production zone[15]. Near the wall, the turbulence intensity increases with the distance from the wall(y). Experiments show that in the viscous sublayer, the longitudinal turbulence intensity increases linearly with y[16], reaching its maximum value within the range of y = 10–20[17,18]. In the absence of secondary flow, the turbulence intensity in all directions decreases with the distance from the wall in the outer region, which can be expressed by an exponential law[18]. When considering the effects of secondary flow, the free surface in the open channel suppresses the vertical fluctuating velocity, and this suppression effect decreases with the distance from the water surface. As a result, the vertical turbulence intensity near the water surface decreases, while the longitudinal and lateral turbulence intensities increase due to the redistribution of turbulence energy[19], which promotes the generation of secondary flow. It is important to note that the above discussion regarding the influence of the water surface on turbulence intensity is based on conditions where the Froude number is not large or the surface tension is sufficiently high to suppress water surface fluctuations. When surface fluctuations are present and the Froude number approaches 1, the turbulence intensity near the water surface increases[16]. Theoretically, it seems more reasonable to expect that the turbulence energy near the water surface should decrease, as large vortices near the surface collide with the water surface, causing deformation of the vortex core and suppressing the vertical turbulence scale[20]. The experimental results of Nakagawa et al.[17]also support this analysis. However, the experiments of Komori et al.[19] show a slight increase in turbulence energy near the water surface. Therefore, the variation of turbulence energy near the water surface remains inconclusive.

Mitigating or eliminating the Karman Vortex Street phenomenon plays an important role in the safe and efficient operation of the project [21–24].

Based on the idea of symmetry with the twist surface of the transition section, this paper proposes a twist surface scheme for the pier type setting scheme and conducts research around the effect.

## 2. Establishment of mathematical model

### 2.1 Model layout

The mathematical model is developed based on the prototype of the Turuohe Aqueduct, a key component of the South-to-North Water Diversion Project.The total length of a large aqueduct project is 660m, the design flow is 350 m³/s, the design water level at the inlet is 146.801m, and the design water level at the outlet is 146.491m; The increased flow is 420 m³/s, the increased water level at the inlet is 147.561m, and the increased water level at the outlet is 147.211m. The available head for design flow is 0.31m. The parameters connecting the upstream channel of the aqueduct are: the top elevation of the bottom plate is 138.801m, the bottom width is 19m, the internal slope is 1:2, and the longitudinal slope i = 1/25000; The parameters of the downstream channel are: the top elevation of the bottom plate is 138.491m, the bottom width is 19m, the internal slope is 1:2, and the longitudinal slope i = 1/25000.

The entrance and exit transition sections of the aqueduct feature a twisted surface design.A twisted surface is a three-dimensional shape that curves or twists in a non-planar manner. Rhino is a powerful professional 3D modeling software developed by Robert McNeel & Assoc in the United States for PCs. Rhino is preferred over other software because it works with NURBS (Non-Uniform Ration B-spline), mathematical representations of 3D geometry that can accurately describe any shape, from the simplest 2D line, circle, arc, or curve to the most complex organic freeform surfaces or solids[25]. Rhino software is used to create the three-dimensional computational model of the aqueduct. In consideration of the influence of inlet and outlet boundary conditions on the numerical simulation results, 200m trapezoidal channel sections are set up at the upstream and downstream of the inlet and outlet transition sections of the aqueduct.

The three-dimensional model of the aqueduct is constructed at a 1:1 scale. The coordinate system is defined as follows: the X-axis represents the lateral direction with the right bank as the positive direction; the Z-axis corresponds to water depth, where the negative direction aligns with gravitational acceleration; and the Y-axis extends along the flow direction, with positive values indicating downstream movement. As shown in Figs 2 and 3, the

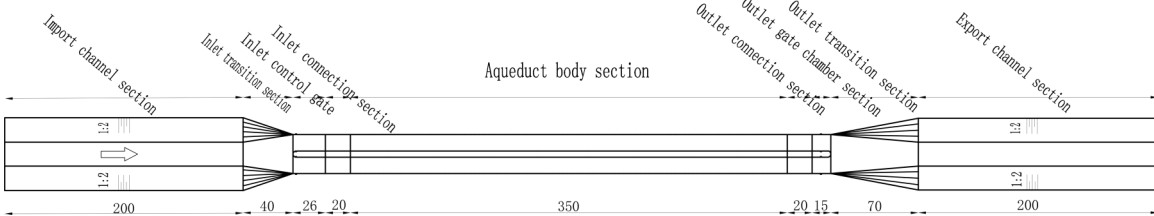

**Fig 2. Aqueduct floor plan.**

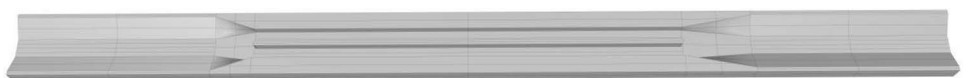

**Fig 3. 3D layout of aqueduct.**

computational model extends 941 m in total length, comprising a 541 m aqueduct section and 200 m of inlet and outlet channels.The cross-section structure of the aqueduct here helps to present the structure of the aqueduct more clearly.

## 2.2 Governing equation

The governing equations are based on the Navier-Stokes equations, which are solved using the Reynolds-averaged approach to account for turbulent effects[26].
Continuity equation:

$$\frac{\partial}{\partial x_i}\left(u_i A_i\right) = 0 \tag{1}$$

Momentum equation:

$$\frac{\partial u_i}{\partial t} + \frac{1}{V_F}\left(\sum_{j=1}^{3} u_j A_j \frac{\partial u_i}{\partial x_j}\right) = -\frac{1}{\rho}\frac{\partial P}{\partial x_i} + g_i + f_i \tag{2}$$

Where i、j = 1, 2, 3; $u_i$ is the direction speed of X、Y、Z. $A_x$、$A_y$、$A_z$ is the area of calculation unit of direction; $V_F$ is the volume fraction of water in each calculation unit; ρ is the density of water;P is the pressure; $g_i$ is gravity; $f_i$ is Reynolds stress [27,28].
    Turbulence model:
The RNG k-ε model is employed due to its effectiveness in handling high streamline curvature and complex flow structures.[29,30].
    Turbulent kinetic energy equation:

$$\frac{\partial(\rho k)}{\partial t} + \frac{\partial(\rho u_i k)}{\partial x_i} = \frac{\partial}{\partial x_j}\left((\mu + \mu_t)\alpha_k \frac{\partial k}{\partial x_j}\right) + G_k - \rho\varepsilon \tag{3}$$

Turbulent energy dissipation rate equation:

$$\frac{\partial(\rho\varepsilon)}{\partial t} + \frac{\partial(\rho u_i \varepsilon)}{\partial x_i} = \frac{\partial}{\partial x_j}\left((\mu + \mu_t)\alpha_\varepsilon \frac{\partial\varepsilon}{\partial x_j}\right) + C_{1\varepsilon}^* \frac{\varepsilon}{k} G_k - C_{2\varepsilon}\rho\frac{\varepsilon^2}{k} \tag{4}$$

Where k is turbulent kinetic energy, $m^2/s^2$; $\varepsilon$ is turbulent energy dissipation rate, $kg \cdot m^2/s^2$; μ is hydrodynamic viscosity coefficient, $N \cdot s/m^2$; $\mu_t$ is fluid turbulent viscosity, $\mu_t = \rho C_\mu k^2/\varepsilon$, $N \cdot s/m^2$; $\alpha_\varepsilon$、$\alpha_k$、$C_{1\varepsilon}$ and B are constants, $\alpha_\varepsilon = \alpha_k = 1.39$, $C_{1\varepsilon}^* = C_{1\varepsilon} - \eta(1 - \eta/\eta_0)/(1 + \beta\eta^3)$, $\eta = (2E_{ij}E_{ij})^{0.5} k/\varepsilon$, $E_{ij} = 1/2(\partial u_i/\partial x_j + \partial u_j/\partial x_i)$, $\eta_0 = 4.337$, $\beta = 0.012$, $C_{1\varepsilon} = 1.42$;constant $C_{2\varepsilon} = 1.68$; $G_k$ is the turbulent kinetic energy generation term caused by the average velocity gradient, $G_k = \mu_t(\partial u_i/\partial u_j + \partial u_j/\partial u_i)\partial u_i/\partial u_j$ [31].

## 2.3 Model meshing

Hexahedral mesh is generated for the model. The quality of grid division affects the accuracy of numerical model calculation results. In order to improve the accuracy of numerical simulation calculation results and consider the calculation time, while ensuring result accuracy, the grid is optimized to reduce computational costs.

A non-uniform grid approach is employed, with refined meshing applied to critical regions to enhance computational accuracy. The study primarily focuses on the hydraulic characteristics of the inlet and outlet transition sections. Nested grid processing is carried out for the positions of the entrance and exit transition sections and the vicinity of the middle pier of the exit transition section. The nested grid of the model exit transition section area is shown in Fig 4(a). The grid size is 0.5m; The overall mesh size is 0.8m; The model consists of approximately 1.395 million computational cells, as illustrated in Fig 4.

### 2.4  Setting of model boundary conditions and initial conditions

**(1)  Boundary condition setting.**  The upstream inlet boundary condition of the model is set as the flow inlet boundary; The downstream outlet boundary condition of the model is set as the pressure outlet boundary, and the water level corresponding to each simulated working condition is given; The wall boundary is given at the left and right banks (positive and negative directions of X axis) and at the bottom of the model (negative direction of Z axis); The pressure outlet is given at the top of the model (positive direction of Z-axis), where the fluid fraction is set to 0, that is, the top of the model is atmospheric pressure.

**(2)  Initial condition setting.**  It can be seen from Fig 2 that the total length of the calculation model is 941m. If the initial conditions of the model do not give the corresponding initial water level according to the simulation conditions, and the tank runs empty, the calculation time will be increased; If only the corresponding initial water level is given, the flow in the flume will vibrate during simulation, and the stability of the model calculation will take a long time. Therefore, on the basis of given specific water level, an average flow rate of the initial water body will be given according to the simulation conditions to reduce the vibration and improve the calculation efficiency. The model unit is set to SI International System of Units, the fluid is selected as 20C liquid water, and the gravity acceleration is set to 9.81m/s$^2$ in the negative direction of Z-axis. In order to clearly and accurately know the stability of the model calculation, monitoring sections are set up at the upstream and downstream of the inlet and outlet to observe the flow and velocity changes of specific sections. When the difference between the instantaneous changes of flow is very small, the model calculation is stable[32].

## 3.  Model validation

### 3.1  Comparison of measured data of model calculation results

A typical aqueduct was measured at 11:00 ~ 14:00 on August 29, 2019. Instantaneous flow is 227.68m³/ s. The water level on the left bank in front of the gate is 147.00m, and the water depth is 8.20m; The water level on the right bank in front of the gate is 146.90m, and the water

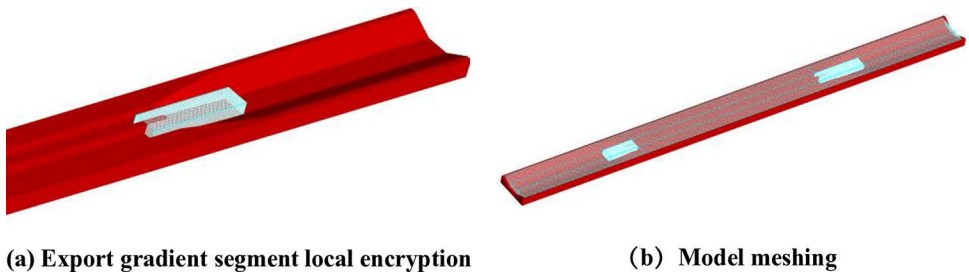

**(a)  Export gradient segment local encryption**          **(b)  Model meshing**

Fig 4.  Grid division.

depth is 8.18m. When measuring the velocity of the aqueduct on site, the first traffic bridge section of the aqueduct along the water flow direction is section A, and the third traffic bridge section is section B. Measuring points are arranged along the right side of section A, one measuring point is arranged every 1m, 12 measuring points are arranged in each tank body, and a total of 24 measuring points are arranged in the two tanks. The arrangement of measuring points is shown in Fig 5.

Set simulation parameters according to the analysis of measured data. The measured flow is 227.68m³/s. The inlet flow boundary is set with a flow of 227.68m³/s; The water level at the outlet pressure boundary is 146.33m; The initial water level of the model is 146.33m; The initial velocity is 0.8m/s. Channel roughness is set as 0.014.

It can be seen from the comparison between the measured water depth and the simulated water depth on the left and right banks of the aqueduct in Table 1 and Table 2 that the water depth at different positions in the aqueduct body is fluctuating, indicating that the numerical

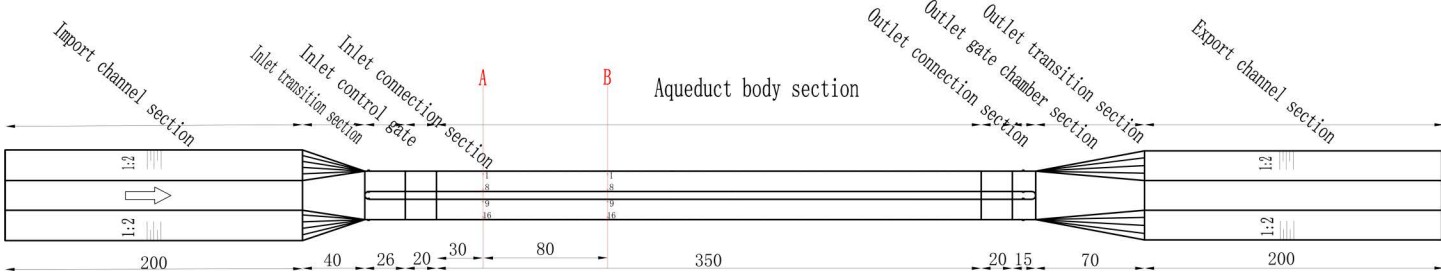

**Fig 5. Layout of measuring points of a typical aqueduct.**

**Table 1. Comparison between measured water depth and simulated water depth on the left bank of aqueduct.**

| Left bank section of aqueduct body section | | | | | | |
|---|---|---|---|---|---|---|
| Measuring point | L1 | L2 | L3 | L4 | L5 | L6 |
| Measured water depth | 5.63 | 5.7 | 5.78 | 5.71 | 5.62 | 5.77 |
| Simulated water depth | 5.58 | 5.68 | 5.75 | 5.75 | 5.67 | 5.71 |
| Measuring point | L7 | L8 | L9 | L10 | L11 | L12 |
| Measured water depth | 5.64 | 5.78 | 5.74 | 5.77 | 5.63 | 5.7 |
| Simulated water depth | 5.58 | 5.73 | 5.68 | 5.71 | 5.72 | 5.76 |

Note: Unit (m)

**Table 2. Comparison between measured water depth and simulated water depth on the right bank of aqueduct.**

| Right bank section of aqueduct body section | | | | | | |
|---|---|---|---|---|---|---|
| Measuring point | R1 | R2 | R3 | R4 | R5 | R6 |
| Measured water depth | 5.65 | 5.77 | 5.80 | 5.71 | 5.74 | 5.65 |
| Simulated water depth | 5.6 | 5.7 | 5.74 | 5.76 | 5.69 | 5.61 |
| Measuring point | R7 | R8 | R9 | R10 | R11 | R12 |
| Measured water depth | 5.79 | 5.66 | 5.74 | 5.80 | 5.64 | 5.77 |
| Simulated water depth | 5.73 | 5.62 | 5.68 | 5.75 | 5.7 | 5.69 |

Note: Unit (m)

simulation can really reflect the actual situation. The maximum difference of water depth is 0.07m, and the percentage of error in measured water depth is 1.2%, which is consistent with the trend of measured water depth data. By comparing the measured water depth and velocity with the numerical simulation, it can be seen that the parameter values and the numerical model establishment and solution results are reliable.

## 3.2 Setting of simulated conditions

It is known from previous studies that Karman Vortex Street is the source of water surface fluctuation. It can be seen from the layout plan of the aqueduct that the tail pier is at the junction of the outlet control gate and the outlet transition section. The downstream of the tail pier is the exit transition section, and there is a bottom slope of 2.207/70 at the bottom of the transition section. Most of the tail piers that have been studied to add extension are flat or streamlined, without considering the impact of the left and right banks. Both the left and right banks are twisted surfaces. In order to enhance the symmetry between the tail pier and the twisted surface on both banks, and make the outlet flow change more evenly with the twisted surface on both banks, the engineering measure studied in this section is to add pier columns with twisted surfaces on both sides behind the original tail pier, and the tail is connected by cone or prism. The specific three-dimensional figure is is shown in Fig 6. Through three-dimensional simulation, the tail pier model measures with good effect are selected.The operating condition settings are shown in Table 3.

## 4. Analysis of simulation results

It can be seen from Fig 7 that there is a relatively obvious Karman Vortex Street phenomenon at the tail of the prototype aqueduct. Different tail pier configurations exhibit varying degrees of effectiveness in suppressing the Kármán Vortex Street, as compared to the prototype aqueduct. In combination condition 3, Karman Vortex Street phenomenon still exists but is weakened, which has an impact on the water surface fluctuation of the aqueduct body and affects the water delivery flow of the aqueduct.

The left and right sides of conditions 2, 4 and 5 are twisted surfaces, and the tail piers are conical for connection. The velocity distribution is shown in Fig 7. Karman Vortex Street phenomenon has been eliminated, and the velocity distribution is symmetrical. Compared to condition 3, conditions 2, 4 and 5 are more effective in eliminating the Karman Vortex Street.

The streamline distribution at the transition section of the tail pier in the prototype aqueduct under Condition 1 is observed. When the tail pier produces a Karman Vortex Street, the streamlines at the tail pier oscillate laterally. As the water moves downward, the amplitude of the swing gradually decreases, and finally returns to the branch line state. The tail pier is added at the exit transition section. Under condition 4, there is no Karman Vortex Street phenomenon at the tail pier. The streamline at the tail pier is straight and there is no left and right swing.

Analyze the water surface profile along the way, and extract the water level along the way with the bottom elevation of the upstream inlet of 0 m. The streamline distribution diagram of the outlet transition section is shown in Fig 8. It can be seen from the water level changes along the way under each working condition in Fig 9 that under Condition 3, water surface fluctuations decrease but still persist, and the water surface profile of the trough body section rises; Due to the influence of Karman Vortex Street at the tail pier, the outlet water level of the transition section of the frustum shaped tail pier is in disorder. Observe condition 2 and condition 3. The pier column volume of condition 2 in the transition section is smaller than that of condition 3. In the outlet transition section, the discharge section area under condition 2 is larger than that under condition 3.

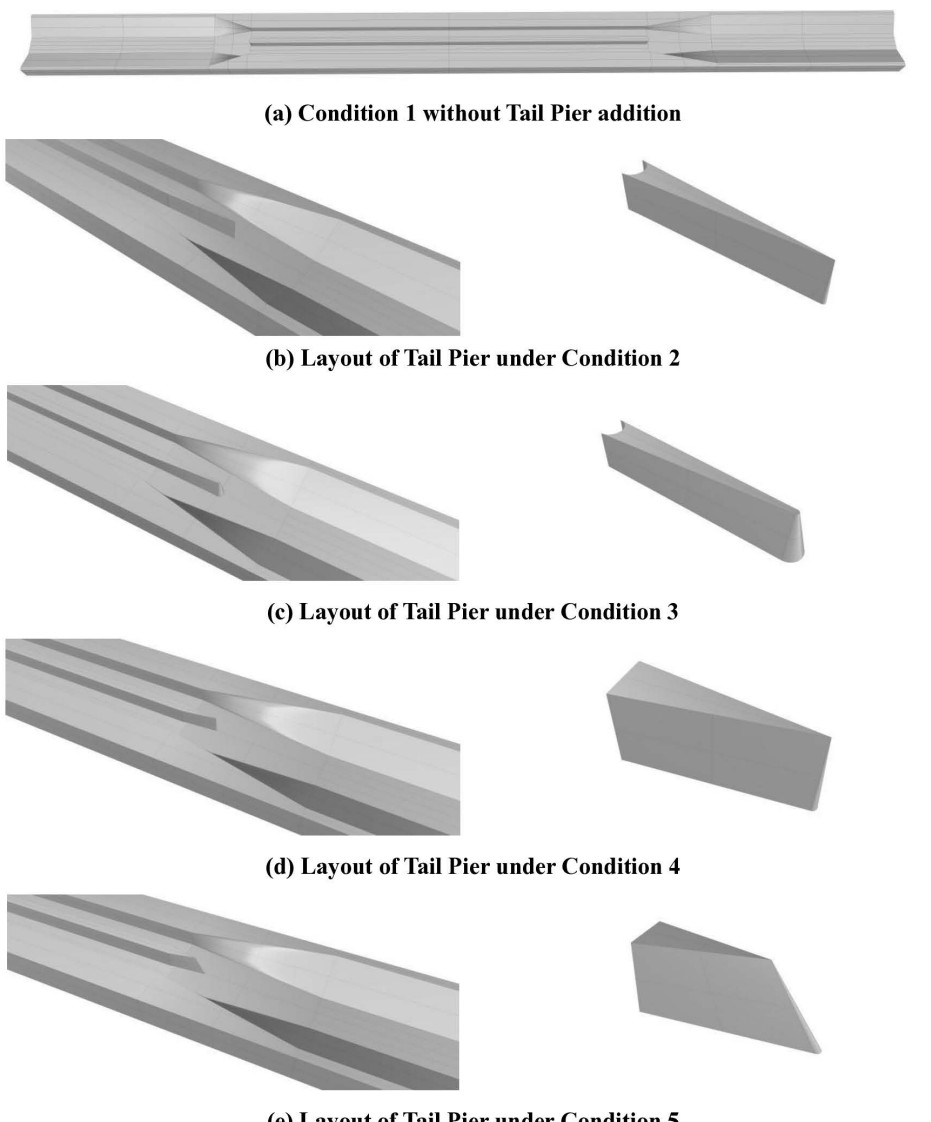

**(a) Condition 1 without Tail Pier addition**

**(b) Layout of Tail Pier under Condition 2**

**(c) Layout of Tail Pier under Condition 3**

**(d) Layout of Tail Pier under Condition 4**

**(e) Layout of Tail Pier under Condition 5**

**Fig 6.  3D Layout Diagrams for Various Conditions.**

When the shape of the entrance transition section is fixed, the width of the middle pier in the trough body affects the water discharge of the aqueduct. The larger the width of the middle pier, the smaller the overall water discharge of the aqueduct will be. In combination condition 3, the cone shaped tail pier has a large water blocking area in the exit transition section, the flow velocity in the transition section decreases, the water level rises, and the water level of the aqueduct body rises compared with the prototype.

It can be seen from the observation of the water level changes along the way under working conditions 4 and 5 that the Karman Vortex Street at the tail pier disappears, there is no reverse transmission of waves in the trench body, the water surface profile of the entire aqueduct is relatively smooth, and there is no water surface fluctuation in the trench body. The analysis of condition 2 and condition 4 shows that both can eliminate the Karman Vortex Street phenomenon, and there is no reverse transmission of waves in

**Table 3. Operating Condition Description and Corresponding figures.**

| | Condition Description | Figure |
|---|---|---|
| Condition 1 | There is no tail pier added. | Fig 6(a) |
| Condition 2 | The tail pier is 35m long, the upper and lower bottom width of the upstream starting end is 5m, the downstream bottom radius is 0.5m, and the upper and lower bottom width is 0m. | Fig 6(b) |
| Condition 3 | The tail pier is 35m long, the upper and lower bottom width of the upstream starting end is 5m, the downstream bottom radius is 2.5m, and the top radius is 0.5m. | Fig 6(c) |
| Condition 4 | The tail pier is 15m long, the upper and lower bottom width of the upstream starting end is 5m, the downstream bottom radius is 0.5m, and the upper and lower bottom width is 0m. | Fig 6(d) |
| Condition 5 | The tail pier bottom is 15m long, the upper part is 15m long, the downstream bottom radius is 0.5m, and the upper bottom width is 0m. | Fig 6(e) |

the aqueduct, so the water surface of the aqueduct is relatively smooth. It can be seen from the observation of the water level along the way that when the tail pier type is the same, the water level along the aqueduct will rise with the increase of the upstream and downstream length of the tail pier. Therefore, the longer the tail pier is, the greater the backwater blocking effect is. Conditions 4 and 5 exhibit similar water surface profiles with no significant differences.

According to the analysis of Froude number along the way, no matter what type of tail pier is added to the exit transition section, the Froude number of trapezoidal channel section at the upstream of the transition section at the entrance of the aqueduct is a relatively smooth curve, which is about 0.15. Karman Vortex Street has no obvious influence on the upstream hydraulic characteristics of the entrance transition section. Due to the gradual contraction of the section at the inlet transition section, the discharge velocity gradually increases, and the Froude number has a sharp increase process of about 0.32. In the aqueduct body section, it can be seen that both the prototype of condition 1 and condition 3 have Karman Vortex Street, and the Froude number in the aqueduct body section will fluctuate with the fluctuation of the water surface in the aqueduct body section. At the exit transition section, the section area gradually increases, and the Froude number of condition 1 and condition 3 with Karman Vortex Street phenomenon fluctuates greatly along the way, while conditions 2, 4 and 5 with Karman Vortex Street elimination are relatively smooth. Fig 10 shows the Froude number distribution.

The turbulent energy dissipation rate[33] quantifies the conversion of kinetic energy into heat due to viscous effects within the flow. Fig 11 shows the turbulent energy dissipation rate distribution. The greater the turbulent energy dissipation rate of fluid, the stronger the intermolecular movement and the stronger the mixing effect. It can be inferred from the turbulence kinetic energy dissipation rates of various model conditions that, for the model conditions involving Karman Vortex Street phenomena, the turbulence kinetic energy dissipation rate shows a significant increase in the outlet transition section, rising from 0.015 to 0.04. Due to the strong water blocking effect of condition 3, the water level in the tank body section is higher than that of the prototype, the cross section velocity of the tank body is smaller, the turbulent energy dissipation rate is smaller than that of the prototype, and the turbulent kinetic energy in the exit transition section increases from 0.001 to 0.0075.

It can be seen from the above research that working conditions 3, 4 and 5 can eliminate the Karman Vortex Street generated by the tail pier. Under these three conditions, both the Froude number and the turbulent energy dissipation rate exhibit smooth variations with no significant differences. Fig 12 shows the total head of the aqueduct under various working conditions. The 15-meter conical tail pier exhibits the lowest head loss and

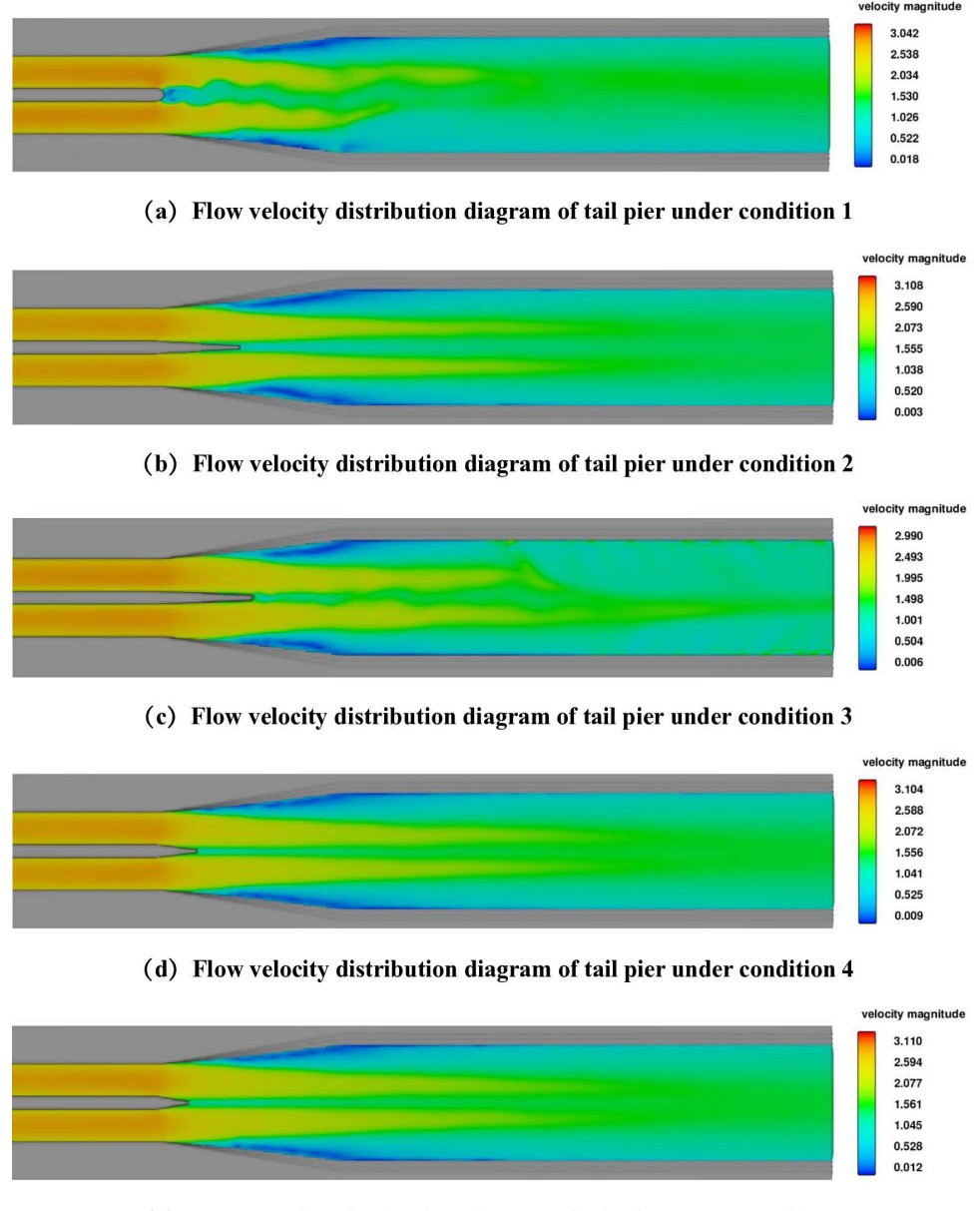

**(a) Flow velocity distribution diagram of tail pier under condition 1**

**(b) Flow velocity distribution diagram of tail pier under condition 2**

**(c) Flow velocity distribution diagram of tail pier under condition 3**

**(d) Flow velocity distribution diagram of tail pier under condition 4**

**(e) Flow velocity distribution diagram of tail pier under condition 5**

**Fig 7. Velocity Distribution of Tail Pier under Condition.**

highest total head, while also being compact, lightweight, and energy-efficient. In comprehensive consideration, conical tail piers with short upstream and downstream lengths are preferred.

After analyzing each tail pier conditional, the platform shaped tail pier can not eliminate the Karman Vortex Street, and the conical tail pier can eliminate the Karman Vortex Street. The water level along the conical tail pier is relatively smooth, and there is no fluctuation of water level in the groove body; The Froude number remains relatively stable within the aqueduct body; The turbulent energy dissipation rate near the exit transition section is obviously smaller than that of the prototype aqueduct. In addition, the 15-meter conical tail pier exhibits

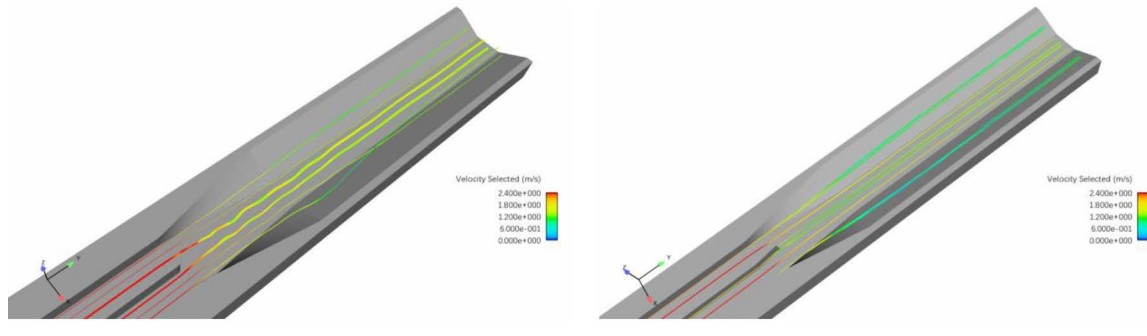

**（a）Streamline Distribution of Tail Pier Containing Karman Vortex Street**

**（b）Streamline Distribution of Tail Pier without Karman Vortex Street**

**Fig 8. Streamline Distribution of Exit Transition Section.**

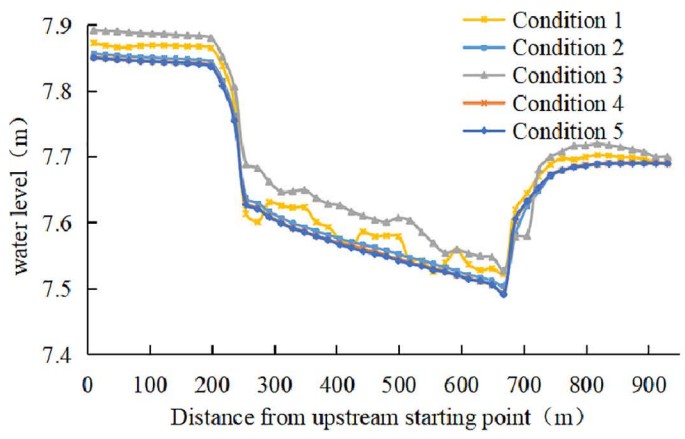
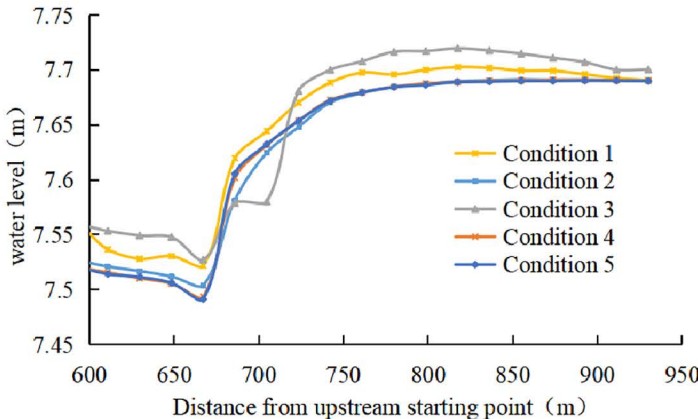

**（a）Water level along the aqueduct**

**（b）Local water level along the transition section of aqueduct outlet**

**Fig 9. Aqueduct Water Level Distribution.**

the least head loss in the exit transition section. The 15m long conical twisted surface tail pier has a good effect.

## 5. Conclusions

In this paper, the following conclusions are drawn through the three-dimensional simulation research on the types of tail piers added in the transition section of the typical aqueduct outlet:

(1) The impact of the Karman Vortex Street is more pronounced in the body section of the channel, as well as in the transition section near the outlet, where water surface fluctuations are more significant. In contrast, the fluctuations in the transition section at the inlet and the upstream trapezoidal channel section are almost negligible.

(2) In the inlet transition section, the discharge section of the section contraction section decreases, the water level drops, and the Froude number and turbulent energy dissipation

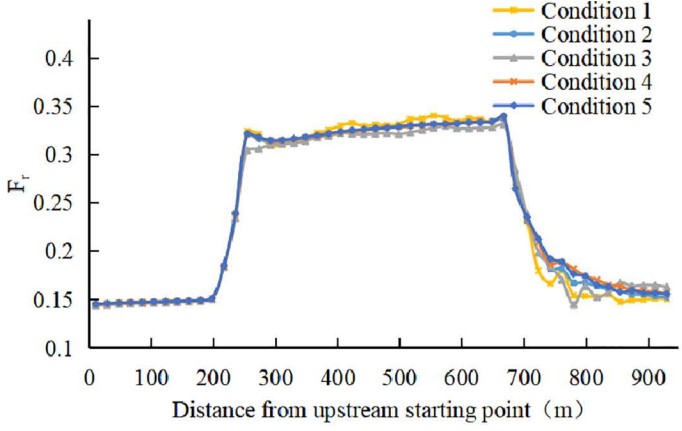

(a)Froude number change along the aqueduct

(b)Local Froude number change at transition section of aqueduct outlet

**Fig 10. Froude number distribution.**

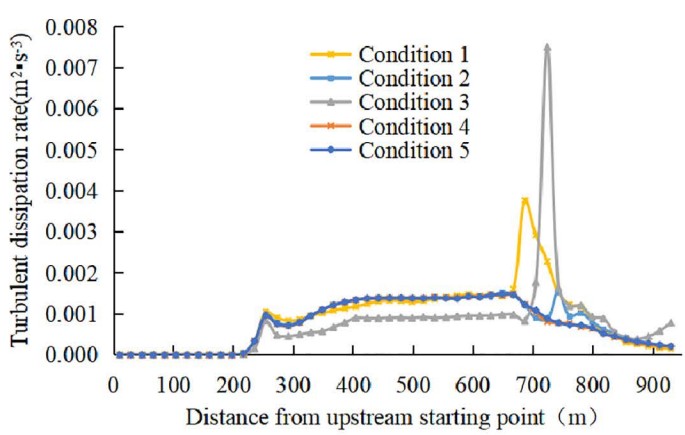 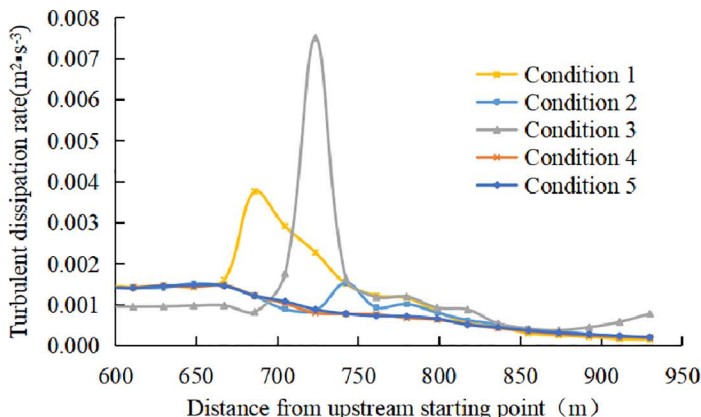

(a)Dissipation rate of turbulent energy consumption along the aqueduct

(b)Dissipation rate of local turbulent energy consumption at transition section of aqueduct outlet

**Fig 11. Turbulent energy dissipation rate distribution.**

rate have a gradual increase process. The discharge section of the outlet transition section gradually increases, the water level rises, and the Froude number and turbulent energy dissipation rate have a gradual reduction process. The Karman Vortex Street at the tail pier significantly increases the turbulent energy dissipation rate in the exit transition section.

(3) Conical tail piers effectively eliminate the Karman Vortex Street, and shorter upstream and downstream lengths result in minimal head loss. The turbulent energy dissipation rate and Froude number are also reduced, resulting in more stable water flow, thus improving the overall hydraulic performance.

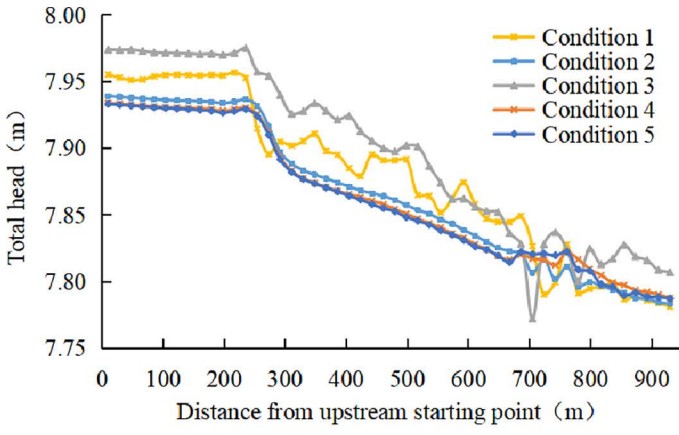

**(a)Total head along the aqueduct**

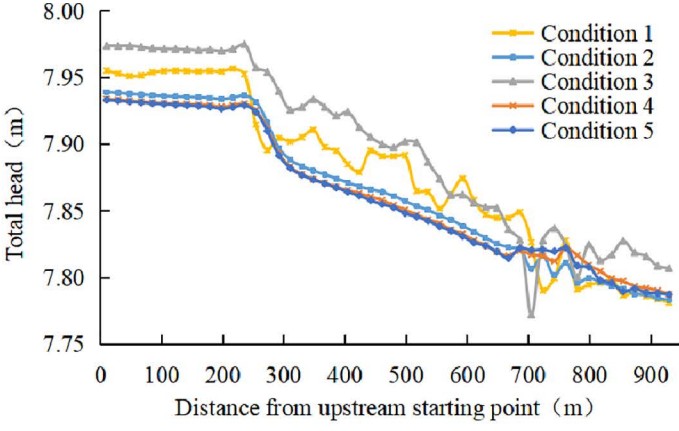

**(b)Local total head of transition section at aqueduct outlet**

**Fig 12. Distribution of Total Head.**

## Author contributions

**Conceptualization:** Hongling Shi.

**Methodology:** Huijie Zhang.

**Writing – original draft:** Jian Chen.

**Writing – review & editing:** Yangyang Tian.

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
