## [Decision Letter · Decision Letter 0]

7 May 2024

PONE-D-23-24085Optimization Measures for Mitigating Water Surface Fluctuations in Large-Scale Water Conveyance AqueductsPLOS ONE

Dear Dr. Shi,

Thank you for submitting your manuscript to PLOS ONE. After careful consideration, we feel that it has merit but does not fully meet PLOS ONE’s publication criteria as it currently stands. Therefore, we invite you to submit a revised version of the manuscript that addresses the points raised during the review process.

We look forward to receiving your revised manuscript.

Kind regards,

Auroop R Ganguly

Academic Editor

PLOS ONE

Promila B (Newgen) 01 Aug 2023: ***Straive, at PRTC, please send the following request. At RTC, please follow usual checks for the observational and field studies: 

"In your Methods section, please provide additional information regarding the permits you obtained for the work. Please ensure you have included the full name of the authority that approved the field site access and, if no permits were required, a brief statement explaining why.

4. We note that your Data Availability Statement is currently as follows: [All relevant data are within the manuscript and its Supporting Information files]

6. Please remove your figures from within your manuscript file, leaving only the individual TIFF/EPS image files, uploaded separately. These will be automatically included in the reviewers’ PDF.

Additional Editor Comments:

The two reviewers have provided detailed comments that require no further amplification from me. While Reviewer 1 recommends minor revision, the comments ranging from presentation style to the need for more comprehensive literature review and detailed discussions may require a major revision. Reviewer 2, however, recommends a rejection and points to deficiencies in the core of the content. Based on the two reviews, I would like to provide the authors one more opportunity through a major revision. I would urge the authors to respond to each and every reviewer comment, either through constructive rebuttals or by performing additional work or both.

Reviewers' comments:

Reviewer's Responses to Questions

**Comments to the Author**

1. Is the manuscript technically sound, and do the data support the conclusions?

Reviewer #1: Yes

Reviewer #2: No

2. Has the statistical analysis been performed appropriately and rigorously? 

Reviewer #1: Yes

Reviewer #2: No

3. Have the authors made all data underlying the findings in their manuscript fully available?

Reviewer #1: Yes

Reviewer #2: No

4. Is the manuscript presented in an intelligible fashion and written in standard English?

Reviewer #1: No

Reviewer #2: Yes

5. Review Comments to the Author

Reviewer #1: In this article, the authors reported a typical three-dimensional hydrodynamic model with optimization of the tail pier structure. The simulation results demonstrate that a conical tail pier can effectively eliminate water surface fluctuations, reduce turbulent energy dissipation rates at the exit transition section, and maintain a relatively stable Froude number and flow pattern. In addition, a 15 m long tapered tail pier incurred less head loss with a small tapered tail pier. The research reported in this article is technically relevant and has certain practical application value.

However, the paper is not well drafted. The authors are required to incorporate the following corrections/clarifications before the manuscript can be considered for publication.

1.Abstract: The abstract needs to do a better job of clarifying and justifying the significance of this study. What is the importance of doing this, and what is the novelty of this work?

2.Introduction: The authors do not provide a good bibliographical review of the current state of the art in this type of problem. We suggest that the authors conduct a good review and place the article in an international context. Please add some recent references in this context (the reference list only contain 20 articles). You need to better signify the importance of the problem and highlight the issues and their significance. The statement of aims is clear, but before this you need to add a gap analysis paragraph that justifies the need for your research.

3.Analysis of simulation results: Results and discussion suggestions are presented separately, and subheadings can be set for discussions under different working conditions.

4.Figures: The quality of all the figures needs improvement, and the target structure should be highlighted.

5.References: Please ensure that all references are relevant to the contents of the manuscript and that suitable recent references are added.

6.Language: Please review the English level and decide whether revisions are necessary.

Reviewer #2: This work presents results from a study examining how tail pier structure could affect water level, energy loss, and water surface fluctuations in aqueducts. The authors test 4 different pier structures and compare the results to an assumed baseline pier shape to determine if there is a reduction in fluctuations. I recommend that the paper be rejected due to the limited scope, overstating of results, and lack of clarity in assumptions and methods. I have included some comments below that may help the authors in refining the content if they choose to modify and resubmit this paper.

Major comments

1. The authors do not clearly define several very important terms that appear repeatedly in the paper, e.g., Karman vortex street (KVS), tail pier, twisted surface. I recommend explaining these terms in the Introduction. It would also be beneficial to briefly describe the cross-section structure of the aqueduct shown in Figs. 2 and 5.

2. The authors should present more instances of KVS occurring in aqueducts.

3. Lines 34-36: the design maximum flow is provided out of context. Has KVS occurred in this system? Has this system observed flows greater than 420 m3/s?

4. Lines 75-77: Why is it important to look at a transition section with a twisted surface? Please provide more clear terminology. It would be useful to include an outline here of what will follow in the paper, i.e., the contents of each section and a brief description of your methodology, results, and metrics of comparison. As it stands, the flow of the paper is hard to follow.

5. In the Introduction, establish a clear gap in the literature that you are trying to address with this work. The authors begin to do this only in Section 3.2 (lines 205-207) but do not clearly explain what a "twisted surface" is or why it is important to model that.

6. Figs. 2 and 5 are very unclear. Please improve the resolution of the image and choose a more clear font.

7. In Section 2 you describe a model of an aqueduct while in Section 3 you mention that you validate the model with measurements taken from a "typical aqueduct". Is the model from Section 2 (as well as its parameters and boundary conditions) based on this aqueduct described in Model 3? If that is the case, please specify in Section 2 that you are building the aqueduct model to represent a very specific existing aqueduct. If the model is not based on a specific aqueduct, how did you obtain the boundary conditions?

8. This study only runs simulations for a very specific case (for which initial and boundary conditions are specified). Therefore, any conclusions from these results cannot be generalized or applied to other aqueduct models or flow regimes. Broad statements in the Conclusion describing "excellent performance" are without justification. Much more robust analyses for a wide range of initial conditions, flow conditions, and aqueduct structures should be tested. Additionally, please add text describing future extensions of this work/avenues of research in the Conclusion section.

9. What is the "strong water blocking effect?" (from line 311 and Fig. 14) and what causes it?

10. The PLOS data policy requires that authors must make their data fully available without restriction. Please consider hosting the model, real aqueduct topology and parameters, and code on a public repository such as Github or in the Supplemental Materials.

Minor comments:

1. Please standardize the spelling of Karman vortex street throughout the paper.

2. Fig. 11 is never described in the text.

3. Figs. 8 and 9 are incorrectly numbered.

4. Line 96: Was the 3D model really constructed at a scale of 1:1?

5. Fig 5: Fig 5 only shows the locations of Sections A and B, it does not show a clear layout of the measuring points of the model as suggested by the caption.

6. Line 37: What does "slapping and pulling beams" mean?

7. Define CFD, RNG when they first appear in the text.

8. There are too many terms to describe the different tail pier structures - "model condition" (line 215), "working condition (Figs 6-9), "condition", "mode" (Section 4). Please standardize the terminology.

6. PLOS authors have the option to publish the peer review history of their article (what does this mean? ). If published, this will include your full peer review and any attached files.

**Do you want your identity to be public for this peer review?** For information about this choice, including consent withdrawal, please see our Privacy Policy .

Reviewer #1: No

Reviewer #2: No

---

## [Author Response · Author response to Decision Letter 1]

27 Nov 2024

Review Comments to the Author

Reviewer #1: In this article, the authors reported a typical three-dimensional hydrodynamic model with optimization of the tail pier structure. The simulation results demonstrate that a conical tail pier can effectively eliminate water surface fluctuations, reduce turbulent energy dissipation rates at the exit transition section, and maintain a relatively stable Froude number and flow pattern. In addition, a 15 m long tapered tail pier incurred less head loss with a small tapered tail pier. The research reported in this article is technically relevant and has certain practical application value.

However, the paper is not well drafted. The authors are required to incorporate the following corrections/clarifications before the manuscript can be considered for publication.

1.Abstract: The abstract needs to do a better job of clarifying and justifying the significance of this study. What is the importance of doing this, and what is the novelty of this work?

Modifications have been made accordingly.

2.Introduction: The authors do not provide a good bibliographical review of the current state of the art in this type of problem. We suggest that the authors conduct a good review and place the article in an international context. Please add some recent references in this context (the reference list only contain 20 articles). You need to better signify the importance of the problem and highlight the issues and their significance. The statement of aims is clear, but before this you need to add a gap analysis paragraph that justifies the need for your research.

Modifications have been made accordingly.

3.Analysis of simulation results: Results and discussion suggestions are presented separately, and subheadings can be set for discussions under different working conditions.

Modifications have been made accordingly.

4.Figures: The quality of all the figures needs improvement, and the target structure should be highlighted.

Modifications have been made accordingly.

5.References: Please ensure that all references are relevant to the contents of the manuscript and that suitable recent references are added.

Modifications have been made accordingly.

6.Language: Please review the English level and decide whether revisions are necessary.

Modifications have been made accordingly.

Reviewer #2: This work presents results from a study examining how tail pier structure could affect water level, energy loss, and water surface fluctuations in aqueducts. The authors test 4 different pier structures and compare the results to an assumed baseline pier shape to determine if there is a reduction in fluctuations. I recommend that the paper be rejected due to the limited scope, overstating of results, and lack of clarity in assumptions and methods. I have included some comments below that may help the authors in refining the content if they choose to modify and resubmit this paper.

Major comments

1.The authors do not clearly define several very important terms that appear repeatedly in the paper, e.g., Karman vortex street (KVS), tail pier, twisted surface. I recommend explaining these terms in the Introduction.It would also be beneficial to briefly describe the cross-section structure of the aqueduct shown in Figs. 2 and 5.

It has been introduced accordingly. A brief description has been provided.

2. The authors should present more instances of KVS occurring in aqueducts.

Relevant introductions have been added.

3.Lines 34-36: the design maximum flow is provided out of context. Has KVS occurred in this system? Has this system observed flows greater than 420 m3/s?

"KVS has occurred. The observed flow rate is not greater than 420."

4.Lines 75-77: Why is it important to look at a transition section with a twisted surface? Please provide more clear terminology. It would be useful to include an outline here of what will follow in the paper, i.e., the contents of each section and a brief description of your methodology, results, and metrics of comparison. As it stands, the flow of the paper is hard to follow.

A twisted surface can guide the fluid to pass through the transition section more smoothly, reducing fluid separation and vortex generation, thereby lowering energy loss.

By precisely designing the shape and angle of the twisted surface, the fluid flow path can be further optimized, enhancing the overall system efficiency.

5. In the Introduction, establish a clear gap in the literature that you are trying to address with this work. The authors begin to do this only in Section 3.2 (lines 205-207) but do not clearly explain what a "twisted surface" is or why it is important to model that.

"Relevant explanations have been added."

6.Figs. 2 and 5 are very unclear. Please improve the resolution of the image and choose a more clear font.

"The Figs has been replaced with clearer Figs."

7.In Section 2 you describe a model of an aqueduct while in Section 3 you mention that you validate the model with measurements taken from a "typical aqueduct". Is the model from Section 2 (as well as its parameters and boundary conditions) based on this aqueduct described in Model 3? If that is the case, please specify in Section 2 that you are building the aqueduct model to represent a very specific existing aqueduct. If the model is not based on a specific aqueduct, how did you obtain the boundary conditions?

The mathematical model is established based on the prototype of the Turuohe Aqueduct in the South-to-North Water Diversion Project.Relevant explanations have been added in Section 2.

8.This study only runs simulations for a very specific case (for which initial and boundary conditions are specified). Therefore, any conclusions from these results cannot be generalized or applied to other aqueduct models or flow regimes. Broad statements in the Conclusion describing "excellent performance" are without justification. Much more robust analyses for a wide range of initial conditions, flow conditions, and aqueduct structures should be tested. Additionally, please add text describing future extensions of this work/avenues of research in the Conclusion section.

This paper aims to explore the impact of different types of tail piers on eliminating the Karman vortex street phenomenon, with a focus on comparing the types of tail piers.

The conclusion section has added possibilities for future research.

9.What is the "strong water blocking effect?" (from line 311 and Fig. 14) and what causes it?

The "strong water blocking effect" (referenced from line 311 and Figure 14) refers to a phenomenon where the flow of water is impeded or restricted, resulting in a reduction in flow velocity, an increase in water level, or a change in flow direction. This effect is typically caused by physical obstacles, narrow passages, sudden changes in pipe diameter, or sharp variations in flow velocity.

In the context of Figure 14 and the related description, the strong water blocking effect may be due to changes in the geometry of the pipeline or container (such as sudden narrowing or widening), internal obstacles, or changes in the angle between the flow direction and the pipe wall. When water flow encounters these changes, it may be obstructed, leading to a decrease in velocity, an increase in water level, and potentially the generation of turbulence or vortices.

10. The PLOS data policy requires that authors must make their data fully available without restriction. Please consider hosting the model, real aqueduct topology and parameters, and code on a public repository such as Github or in the Supplemental Materials.

Minor comments:

1.Please standardize the spelling of Karman vortex street throughout the paper.

It has been processed in accordance with standards.

2.Fig. 11 is never described in the text.

The corresponding instructions have been added.

3.Figs. 8 and 9 are incorrectly numbered.

The change has been made.

4.Line 96: Was the 3D model really constructed at a scale of 1:1?

Yes, it is a mathematical model.

5.Fig 5: Fig 5 only shows the locations of Sections A and B, it does not show a clear layout of the measuring points of the model as suggested by the caption.

The locations of the measurement points are described in sufficient detail in the text.

6.Line 37: What does "slapping and pulling beams" mean?

The article has been revised to: "Slapping against support beams"."Slapping against support beams" refers to the water hitting and impacting the structural beams supporting the aqueduct.

7.Define CFD, RNG when they first appear in the text.

Relevant definitions have been added.

8.There are too many terms to describe the different tail pier structures - "model condition" (line 215), "working condition (Figs 6-9), "condition", "mode" (Section 4). Please standardize the terminology.

The terminology has been standardized.

---

## [Decision Letter · Decision Letter 1]

13 Dec 2024

PONE-D-23-24085R1Study on optimization measures for water level fluctuation of large water conveyance aqueductsPLOS ONE

Dear Dr. Shi,

Thank you for submitting your manuscript to PLOS ONE. After careful consideration, we feel that it has merit but does not fully meet PLOS ONE’s publication criteria as it currently stands. Therefore, we invite you to submit a revised version of the manuscript that addresses the points raised during the review process.

We look forward to receiving your revised manuscript.

Kind regards,

Auroop R Ganguly

Academic Editor

PLOS ONE

Additional Editor Comments :

Reviewer 1 had relatively minor comments, but the comments from Reviewer 2 are clear and many underlying issues remain unresolved in the revised version. The authors may want to focus on the new comments from Reviewer 2. The comments are clear and require no further amplification from me, but I would recommend the authors take them very seriously and address (and/or if applicable, rebut) them. A strong response to the comments from Reviewer 2 is a prerequisite for acceptance.

Reviewers' comments:

Reviewer's Responses to Questions

**Comments to the Author**

1. If the authors have adequately addressed your comments raised in a previous round of review and you feel that this manuscript is now acceptable for publication, you may indicate that here to bypass the “Comments to the Author” section, enter your conflict of interest statement in the “Confidential to Editor” section, and submit your "Accept" recommendation.

Reviewer #2: (No Response)

2. Is the manuscript technically sound, and do the data support the conclusions?

Reviewer #2: Partly

3. Has the statistical analysis been performed appropriately and rigorously? 

Reviewer #2: I Don't Know

4. Have the authors made all data underlying the findings in their manuscript fully available?

Reviewer #2: No

5. Is the manuscript presented in an intelligible fashion and written in standard English?

Reviewer #2: No

6. Review Comments to the Author

Reviewer #2: In the revised manuscript, I acknowledge the authors' effort to add more information to the Introduction as per the reviewers' past comments. However, it appears to me that the authors have not made many other significant efforts to address the suggestions and guidance provided in the previous round of comments. As an example, even though the authors have responded to comments from both reviewers stating that modifications have been made (regarding adding more references to the paper and standardizing the spelling of the core term "Karman Vortex Street"), it seems that these changes were not actually made. No additional references were added and spelling inconsistencies are still present throughout the paper. Additionally, I was unable to view any tracked changes in the revised manuscript. I strongly encourage the authors to very carefully go through the paper and make revisions to make this paper worthy of publication.

1. In the previous round of revisions, the authors were asked to standardize the spelling and capitalization of the critical "Karman Vortex Street" term throughout the paper, and while the authors claim to have addressed this issue in their responses, the same errors persist (line 15 in the abstract, line 40, line 95, line 98 ...). Please take care to standardize these spellings.

2. Abstract line 22: "a conical tail pier performs better in eliminating water surface fluctuations": performs better than what other tail pier shapes?

3. Abstract line 26: "a shorter, 15-meter conical tail pier incurs less head loss": less head loss compared to what?

4. The word "Conditional" has been spelled "Conditionl" throughout the paper

6. I suggest reformatting each reference very carefully. Perhaps consider using a reference management software to standardize the format. The formatting is very inconsistent and the references themselves are hard to track and sometimes wrong - for example, the first paper should be from "Chinese Journal of Nature" and not "Nature Journal."

7. I think the introduction needs to be made much stronger and the authors should be very careful and clear with the details. For example, the "Tacoma Canyon Bridge" in reality is named the "Tacoma Narrows Bridge." Also, the current format of the introduction is difficult to follow - lines 56-62 are not connected to the KVS specifically. And in lines 71 and 79 - what is the "numerical simulation conditional" and the "physical conditional test"? You use the term "conditional" throughout the paper but I do not see a clear explanation of what that means. Finally, I think adding more studies, as the other reviewer suggested, is critical to improve the trustworthiness and quality of this paper.

8. Line 56: "South to North Water Transfer Project" located where? Please remember that PLOS One has an international readership.

9. Line 139: N-S is Navier Stokes equation?

10. In Table 1 and 2 - please add the unit of measure for measured and simulated water depth ([m] I presume)

12. Line 119: add a citation for Rhino software

13. The paper contains two many modeling specifics (e.g., lines 107 - 115, line 175- 178, lines 220 - 225, etc.) that should be instead placed in a table or in the Supplmental Materials. They do not need to be in the body of the manuscript.

14. Section 3 Conditional Validation: where is the description and figure of condition 1?

15. I suggest combining Figs 6 - 9 into into one figure and replacing the text describing each condition (lines 255 - 269) with a table containing a summary of this information.

16. Would it be possible to use a different color scheme for Fig 10 to clearly show the KVS effects? It is a little unclear now since the yellow, blue, and green are not easy to differentiate.

17. The authors have not addressed my previous comment: The PLOS data policy requires that authors must make their data fully available without restriction. Please consider hosting the model, real aqueduct topology and parameters, and code on a public repository such as Github or in the Supplemental Materials.

7. PLOS authors have the option to publish the peer review history of their article (what does this mean? ). If published, this will include your full peer review and any attached files.

**Do you want your identity to be public for this peer review?** For information about this choice, including consent withdrawal, please see our Privacy Policy .

Reviewer #2: No

---

## [Author Response · Author response to Decision Letter 2]

26 Dec 2024

1.In the previous round of revisions, the authors were asked to standardize the spelling and capitalization of the critical "Karman Vortex Street" term throughout the paper, and while the authors claim to have addressed this issue in their responses, the same errors persist (line 15 in the abstract, line 40, line 95, line 98 ...). Please take care to standardize these spellings.

The corresponding content has been fully modified

2.Abstract line 22: "a conical tail pier performs better in eliminating water surface fluctuations": performs better than what other tail pier shapes?

The experimental results demonstrate that a conical tail pier performs better than prismy tail pier in eliminating water surface fluctuations.

3.Abstract line 26: "a shorter, 15-meter conical tail pier incurs less head loss": less head loss compared to what?

Furthermore, by comparing the total head loss at the outlet tapering section, it is found that a shorter, 15-meter conical tail pier incurs less head loss than 35-meter conical tail pier, indicating that a smaller conical tail pier is more effective.

4.The word "Conditional" has been spelled "Conditionl" throughout the paper

The corresponding content has been fully modified.

6.I suggest reformatting each reference very carefully. Perhaps consider using a reference management software to standardize the format. The formatting is very inconsistent and the references themselves are hard to track and sometimes wrong - for example, the first paper should be from "Chinese Journal of Nature" and not "Nature Journal."

I have re-formatted each reference, and now they all follow the same format. The references themselves can also be traced.

7.I think the introduction needs to be made much stronger and the authors should be very careful and clear with the details. For example, the "Tacoma Canyon Bridge" in reality is named the "Tacoma Narrows Bridge." Also, the current format of the introduction is difficult to follow - lines 56-62 are not connected to the KVS specifically.And in lines 71 and 79 - what is the "numerical simulation conditional" and the "physical conditional test"?You use the term "conditional" throughout the paper but I do not see a clear explanation of what that means.Finally, I think adding more studies, as the other reviewer suggested, is critical to improve the trustworthiness and quality of this paper.

"Tacoma Canyon Bridge" has been changed to "Tacoma Narrows Bridge."

The content here is intended to introduce the negative impact of the Karman vortex street phenomenon in the water conveyance process of a flume by using a specific engineering example, further illustrating the hazards of the Karman Vortex Streets.The subsequent content of the article also focuses on research based on this flume as the prototype.

These two respectively represent “numerical simulation model” and “physical model experiments”, and I have already made the corresponding modifications in the text.

To avoid misunderstanding, I changed "conditional" to "model"in the necessary places.

"I have added content regarding open-channel turbulence and water conveyance capacity, corresponding to lines 77 to 101 in the text."

The anisotropy of turbulence in open channels is caused by the combined effects of the solid boundary and the water surface. The near-wall region of the open channel is the turbulence production zone [2]. Near the wall, the turbulence intensity increases with the distance from the wall(y). Experiments show that in the viscous sublayer, the longitudinal turbulence intensity increases linearly with y [3], reaching its maximum value within the range of y = 10–20 [4,5]. In the absence of secondary flow, the turbulence intensity in all directions decreases with the distance from the wall in the outer region, which can be expressed by an exponential law [5]. When considering the effects of secondary flow, the free surface in the open channel suppresses the vertical fluctuating velocity, and this suppression effect decreases with the distance from the water surface. As a result, the vertical turbulence intensity near the water surface decreases, while the longitudinal and lateral turbulence intensities increase due to the redistribution of turbulence energy [6], which promotes the generation of secondary flow. It is important to note that the above discussion regarding the influence of the water surface on turbulence intensity is based on conditions where the Froude number is not large or the surface tension is sufficiently high to suppress water surface fluctuations. When surface fluctuations are present and the Froude number approaches 1, the turbulence intensity near the water surface increases[3]. Theoretically, it seems more reasonable to expect that the turbulence energy near the water surface should decrease, as large vortices near the surface collide with the water surface, causing deformation of the vortex core and suppressing the vertical turbulence scale [7]. The experimental results of Nakagawa et al.[4]also support this analysis. However, the experiments of Komori et al. [6] show a slight increase in turbulence energy near the water surface. Therefore, the variation of turbulence energy near the water surface remains inconclusive.

8.Line 56: "South to North Water Transfer Project" located where? Please remember that PLOS One has an international readership.

The relevant explanation has been added：

The Central Line Project of South-to-North Water Diversion draws water from the Danjiangkou Reservoir, passes through Henan and Hubei provinces, crosses the Yellow River, and delivers water northward to Beijing and other northern regions. It is a major water transfer project connecting southern and northern China.

9.Line 139: N-S is Navier Stokes equation?

Yes, I have made the modifications in the text.

10.In Table 1 and 2 - please add the unit of measure for measured and simulated water depth ([m] I presume)

I have added notes to the table.

12.Line 119: add a citation for Rhino software

Rhino is a powerful professional 3D modeling software developed by Robert McNeel & Assoc in the United States for PCs. It can be widely applied in fields such as 3D animation production, industrial manufacturing, scientific research, and mechanical design.

13.The paper contains two many modeling specifics (e.g., lines 107 - 115, line 175- 178, lines 220 - 225, etc.) that should be instead placed in a table or in the Supplmental Materials. They do not need to be in the body of the manuscript.

Thank you for your suggestion. I have removed some of the modeling details from the manuscript's main text and retained some basic explanations.

14.Section 3 Conditional Validation: where is the description and figure of condition 1?

I have added the relevant image and description in the text. Please see Figure 6.

15.I suggest combining Figs 6 - 9 into into one figure and replacing the text describing each condition (lines 255 - 269) with a table containing a summary of this information.

The modifications have been completed as suggested.

16.Would it be possible to use a different color scheme for Fig 10 to clearly show the KVS effects? It is a little unclear now since the yellow, blue, and green are not easy to differentiate.

I'd prefer to keep the current color scheme and have replaced the figure with a clearer one for you to observe more easily.

17. The authors have not addressed my previous comment: The PLOS data policy requires that authors must make their data fully available without restriction. Please consider hosting the model, real aqueduct topology and parameters, and code on a public repository such as Github or in the Supplemental Materials.

I have added all the data and models in the Supplemental Materials.

---

## [Decision Letter · Decision Letter 2]

14 Jan 2025

PONE-D-23-24085R2Study on optimization measures for water level fluctuation of large water conveyance aqueductsPLOS ONE

Dear Dr. Shi,

Thank you for submitting your manuscript to PLOS ONE. After careful consideration, we feel that it has merit but does not fully meet PLOS ONE’s publication criteria as it currently stands. Therefore, we invite you to submit a revised version of the manuscript that addresses the points raised during the review process.

We look forward to receiving your revised manuscript.

Kind regards,

Auroop R Ganguly

Academic Editor

PLOS ONE

Additional Editor Comments :

The authors are requested to make a strong attempt to address the comments from Reviewer 2. The comments are clear and many have persisted across revision cycles, and need no further amplification from me. The acceptance of this manuscript is contingent on addressing the comments from this reviewer.

Reviewers' comments:

Reviewer's Responses to Questions

**Comments to the Author**

1. If the authors have adequately addressed your comments raised in a previous round of review and you feel that this manuscript is now acceptable for publication, you may indicate that here to bypass the “Comments to the Author” section, enter your conflict of interest statement in the “Confidential to Editor” section, and submit your "Accept" recommendation.

Reviewer #2: (No Response)

2. Is the manuscript technically sound, and do the data support the conclusions?

Reviewer #2: Partly

3. Has the statistical analysis been performed appropriately and rigorously? 

Reviewer #2: I Don't Know

4. Have the authors made all data underlying the findings in their manuscript fully available?

Reviewer #2: Yes

5. Is the manuscript presented in an intelligible fashion and written in standard English?

Reviewer #2: No

6. Review Comments to the Author

Reviewer #2: I thank the authors for the quick turn around time and responses to my comments. Please find more comments below.

1. In response to my previous comment: "Abstract line 22: "a conical tail pier performs better in eliminating water surface fluctuations": performs better than what other tail pier shapes?" the authors have changed the line in the abstract to "conical tail pier performs better than prismy tail pier in eliminating water surface fluctuations." Are you sure "prismy" is the right word? It does not seem like standard English and you do not mention the word "prismy" anywhere else in the manuscript.

2. Line 27: why did you choose 15 and 35-meter conical tail pier specifically? Are these common tail pier lengths? Please either generalize the text (e.g., just say short or long) or explain why you chose these numbers.

3. Thank you for adding more references. I think the Introduction is now more comprehensive and explanatory. However, I still recommend finding and adding credible references to strengthen at least the following sentences because they introduce important definitions and examples:

- both the first and second sentences in the Introduction (line 36-38, line 38-40).

- Line 51-56

4. While the Introduction is more comprehensive now, it is still very disjointed. E.g., in lines 52-56, the authors claim that one can apply a number of practical engineering measures to suppress the generation of the Karman Vortex Street. This sentence is immediately followed by what I believe is a description of the authors' testing scenario. Examples of practical engineering measures only appear starting line 102. In the middle of these two sections is the seemingly random placement of a section on studies on open channel turbulence. I stress, just like in my previous round of comments, that the authors should carefully revisit the format of the Introduction. The content is now stronger than it was previously, but this ordering is confusing and hard to follow.

5. Line 151: add a reference for Rhino.

6. Thank you for creating Table 3. I suggest following PLOS One table formatting requirements to make the table a little easier to read and minimize the number of revisions going forward. https://journals.plos.org/plosone/s/tables Adding cell borders will help. You could also make it simpler to read by removing the long sentences in "Condition Description" and instead creating separate columns for length, upstream higher bottom width, upstream lower bottom width, etc. I will leave that up to you.

7. Lines 395-398: I'm afraid I don't understand this wording. Do you mean that the Karman Vortex Street only causes flow fluctuations in the aqueduct body section and exit transition section and has no effect on the other sections?

8. Lines 406-409: I suggest rewording this. "The effect is good" is not suitable language for an academic journal.

9. Line 410-411: please connect these lines back to the Karman Vortex Street.

10. Regarding references: Thank you for improving the references. Again, to minimize further revisions, I suggest that you closely follow the PLOS One reference requirements for every single reference: https://journals.plos.org/plosone/s/submission-guidelines#loc-references You will notice that the required ICMJE reference style requires abbreviated journal names, e.g., Measurement Science and Technology should probably be written as Meas Sci Technol (please double check that). The PLOS One guidelines also specify how to write references for articles not written in English.

11. Since the journal does not copyedit accepted manuscripts, the language should be significantly improved before it can be published. I suggest submitting the paper to an English-speaking third party to improve the grammar and choice of words used throughout the paper.

7. PLOS authors have the option to publish the peer review history of their article (what does this mean? ). If published, this will include your full peer review and any attached files.

**Do you want your identity to be public for this peer review?** For information about this choice, including consent withdrawal, please see our Privacy Policy .

Reviewer #2: No

---

## [Author Response · Author response to Decision Letter 3]

31 Jan 2025

Dear Editor and Reviewer,

We sincerely thank the editor and all reviewers for their valuable feedback that we have used to improve the quality of our manuscript. The reviewer comments are laid out below in italicized font and specific concerns have been numbered. Our response is given in normal font and changes/additions to the manuscript are given in the blue text.

Reviewer 2

Comment 1:In response to my previous comment: "Abstract line 22: "a conical tail pier performs better in eliminating water surface fluctuations": performs better than what other tail pier shapes?" the authors have changed the line in the abstract to "conical tail pier performs better than prismy tail pier in eliminating water surface fluctuations." Are you sure "prismy" is the right word? It does not seem like standard English and you do not mention the word "prismy" anywhere else in the manuscript.

Response:Thank you for your suggestion. I have corrected the mistake and replaced "prismy tail pier" with "platform shaped tail pier."

Comment 2:Line 27: why did you choose 15 and 35-meter conical tail pier specifically? Are these common tail pier lengths? Please either generalize the text (e.g., just say short or long) or explain why you chose these numbers.

Response:The intention here is only to express the difference between relatively longer and relatively shorter, and more appropriate modifications have already been made in the text. The following is the revised content.

Furthermore, by comparing the total head loss at the outlet tapering section, it is found that a shorter（e.g.,15-meter） conical tail pier incurs less head loss than a longer（e.g.,35-meter） conical tail pier, indicating that a smaller conical tail pier is more effective. This suggests that shorter tail piers are generally more effective in reducing head loss and improving flow stability.

Comment 3: Thank you for adding more references. I think the Introduction is now more comprehensive and explanatory. However, I still recommend finding and adding credible references to strengthen at least the following sentences because they introduce important definitions and examples:

- both the first and second sentences in the Introduction (line 36-38, line 38-40).

- Line 51-56

Response:I have added the relevant references to both the first and second sentences in the Introduction and provided a more detailed explanation of the original sentences. The specific content is as follows:

The Karman Vortex Street was first described by von Kármán as a phenomenon where “two parallel rows of vortices alternating in direction are shed in the wake of an obstacle in a fluid with moderate Reynolds numbers.” The wind-induced damage incident of the Tacoma Narrows Bridge in the United States was an accident caused by the Karman Vortex Street. The shedding frequency of the Karman Vortex Street was close to the natural frequency of the bridge deck, leading to resonance. As a result, the amplitude of the vibrations gradually increased, ultimately causing structural instability and collapse.

The original text - Lines 51-56 have also been supplemented with relevant references, namely [4,5,6,7].

Comment 4:While the Introduction is more comprehensive now, it is still very disjointed. E.g., in lines 52-56, the authors claim that one can apply a number of practical engineering measures to suppress the generation of the Karman Vortex Street. This sentence is immediately followed by what I believe is a description of the authors' testing scenario. Examples of practical engineering measures only appear starting line 102. In the middle of these two sections is the seemingly random placement of a section on studies on open channel turbulence. I stress, just like in my previous round of comments, that the authors should carefully revisit the format of the Introduction. The content is now stronger than it was previously, but this ordering is confusing and hard to follow.

Response:Thank you very much for your suggestions; this is an important issue. I have adjusted the order of the relevant content, connecting the practical engineering research more closely with the previous section, and moved the research on open channel turbulence after the practical engineering research. I have also added an explanation of the transition section. The content of the transition section is as follows:

Besides the practical application studies in water conveyance engineering, theoretical analysis of open channel turbulence also provides an important foundation for understanding the flow characteristics of aqueducts. The anisotropy of open channel turbulence and the impact of free surfaces on turbulence intensity are directly related to the flow stability in the transition sections of aqueducts. Therefore, it is necessary to combine the research findings on open channel turbulence to further explore the mechanisms of flow fluctuations in aqueducts.

Comment 5:Line 151: add a reference for Rhino.

Response:I have added a reference for Rhino, and the specific content is as follows.

Rhino is preferred over other software because it works with NURBS (Non-Uniform Ration B-spline), mathematical representations of 3D geometry that can accurately describe any shape, from the simplest 2D line, circle, arc, or curve to the most complex organic freeform surfaces or solids.

Comment 6:Thank you for creating Table 3. I suggest following PLOS One table formatting requirements to make the table a little easier to read and minimize the number of revisions going forward. https://journals.plos.org/plosone/s/tables Adding cell borders will help. You could also make it simpler to read by removing the long sentences in "Condition Description" and instead creating separate columns for length, upstream higher bottom width, upstream lower bottom width, etc. I will leave that up to you.

Response:Thank you for your suggestion. I have modified all the tables in the manuscript according to the formatting requirements of PLOS ONE.

Comment 7:Lines 395-398: I'm afraid I don't understand this wording. Do you mean that the Karman Vortex Street only causes flow fluctuations in the aqueduct body section and exit transition section and has no effect on the other sections?

Response:As you mentioned, the original sentence was unclear. I have made adjustments to avoid ambiguity. The revised content is as follows:

The impact of the Karman Vortex Street is more pronounced in the body section of the channel, as well as in the transition section near the outlet, where water surface fluctuations are more significant. In contrast, the fluctuations in the transition section at the inlet and the upstream trapezoidal channel section are almost negligible.

Comment 8:Lines 406-409: I suggest rewording this. "The effect is good" is not suitable language for an academic journal.

Response:Thank you very much for your suggestion. I have made the revisions, and the revised content is as follows：

Conical tail piers can eliminate Karman Vortex Street, and when the upstream and downstream lengths are short, the head loss is minimal. The turbulent energy dissipation rate and Froude number are also reduced, resulting in more stable water flow, thus improving the overall hydraulic performance.

Comment 9:Line 410-411: please connect these lines back to the Karman Vortex Street.

Response:This section is forward-looking and is not suitable for inclusion in the conclusion. I have removed the corresponding content.

Comment 10:Regarding references: Thank you for improving the references. Again, to minimize further revisions, I suggest that you closely follow the PLOS One reference requirements for every single reference: https://journals.plos.org/plosone/s/submission-guidelines#loc-references You will notice that the required ICMJE reference style requires abbreviated journal names, e.g., Measurement Science and Technology should probably be written as Meas Sci Technol (please double check that). The PLOS One guidelines also specify how to write references for articles not written in English.

Response:Thank you very much for your valuable suggestions. I have carefully revised the reference format in accordance with the guidelines provided by PLOS ONE.

Comment 11:Since the journal does not copyedit accepted manuscripts, the language should be significantly improved before it can be published. I suggest submitting the paper to an English-speaking third party to improve the grammar and choice of words used throughout the paper.

Response:Thank you very much for your suggestions. I have submitted my paper to a native English speaker specializing in hydraulics, and with their assistance, I have significantly enhanced the overall language quality of the manuscript.

---

## [Decision Letter · Decision Letter 3]

19 Feb 2025

Study on optimization measures for water level fluctuation of large water conveyance aqueducts

PONE-D-23-24085R3

Dear Dr. Shi,

We’re pleased to inform you that your manuscript has been judged scientifically suitable for publication and will be formally accepted for publication once it meets all outstanding technical requirements.

Kind regards,

Auroop R Ganguly

Academic Editor

PLOS ONE

Additional Editor Comments (optional):

I am glad the authors have addressed all reviewer comments and would recommend acceptance.

Reviewers' comments:

Reviewer's Responses to Questions

**Comments to the Author**

1. If the authors have adequately addressed your comments raised in a previous round of review and you feel that this manuscript is now acceptable for publication, you may indicate that here to bypass the “Comments to the Author” section, enter your conflict of interest statement in the “Confidential to Editor” section, and submit your "Accept" recommendation.

Reviewer #2: All comments have been addressed

2. Is the manuscript technically sound, and do the data support the conclusions?

Reviewer #2: Yes

3. Has the statistical analysis been performed appropriately and rigorously? 

Reviewer #2: I Don't Know

4. Have the authors made all data underlying the findings in their manuscript fully available?

Reviewer #2: Yes

5. Is the manuscript presented in an intelligible fashion and written in standard English?

Reviewer #2: Yes

6. Review Comments to the Author

Reviewer #2: (No Response)

7. PLOS authors have the option to publish the peer review history of their article (what does this mean? ). If published, this will include your full peer review and any attached files.

**Do you want your identity to be public for this peer review?** For information about this choice, including consent withdrawal, please see our Privacy Policy .

Reviewer #2: No

---

## [Editor Report · Acceptance letter]

PONE-D-23-24085R3

PLOS ONE

Dear Dr. Shi,

I'm pleased to inform you that your manuscript has been deemed suitable for publication in PLOS ONE. Congratulations! Your manuscript is now being handed over to our production team.

Kind regards,

on behalf of

Dr. Auroop R Ganguly

Academic Editor

PLOS ONE